

# The disturbance leg-lift response (DLR): an undescribed behavior in bumble bees

Christopher A. Varnon, Noelle Vallely, Charlie Beheler and Claudia Coffin

Department of Psychology, Converse College, Spartanburg, SC, United States of America

## ABSTRACT

**Background**. Bumble bees, primarily *Bombus impatiens* and *B. terrestris*, are becoming increasingly popular organisms in behavioral ecology and comparative psychology research. Despite growing use in foraging and appetitive conditioning experiments, little attention has been given to innate antipredator responses and their ability to be altered by experience. In this paper, we discuss a primarily undescribed behavior, the disturbance leg-lift response (DLR). When exposed to a presumably threatening stimulus, bumble bees often react by lifting one or multiple legs. We investigated DLR across two experiments.

**Methods**. In our first experiment, we investigated the function of DLR as a prerequisite to later conditioning research. We recorded the occurrence and sequence of DLR, biting and stinging in response to an approaching object that was either presented inside a small, clear apparatus containing a bee, or presented directly outside of the subject's apparatus. In our second experiment, we investigated if DLR could be altered by learning and experience in a similar manner to many other well-known bee behaviors. We specifically investigated habituation learning by repeatedly presenting a mild visual stimulus to samples of captive and wild bees.

**Results**. The results of our first experiment show that DLR and other defensive behaviors occur as a looming object approaches, and that the response is greater when proximity to the object is lower. More importantly, we found that DLR usually occurs first, rarely precedes biting, and often precedes stinging. This suggests that DLR may function as a warning signal that a sting will occur. In our second experiment, we found that DLR can be altered as a function of habituation learning in both captive and wild bees, though the captive sample initially responded more. This suggests that DLR may be a suitable response for many other conditioning experiments.

## INTRODUCTION

The study of the psychological abilities of bees has become an important research area. Such research provides insights to the valuable and global role of bees in agriculture and in the ecosystem. Additionally, bees are also excellent model organisms for investigating the relationships between complex behavior, ecological demands, and neurophysiology, and are the most researched invertebrate in recent comparative psychology (*Varnon, Lang & Abramson, 2018*). Psychological research with bees involves a number of topics including color perception (*Koethe et al., 2016*), olfactory learning (*Riveros & Gronenberg, 2009b*),

Corresponding author
Christopher A. Varnon,
christopher.varnon@converse.edu

perception of time (*Craig et al., 2014*), conditioned taste aversion (*Varnon et al., 2018*), learned helplessness (*Dinges et al., 2017*), select and reject stimulus control (*Scienza et al., 2019*), concept learning (*Giurfa et al., 2001*), social transmission of learned behaviors (*Alem et al., 2016*), acquisition and flexibility of foraging skills (*Raine & Chittka, 2007*; *Strang & Sherry, 2014*), maximization of resources (*Charlton & Houston, 2010*), effects of pesticides on learning (*Stanley, Smith & Raine, 2015*), alcoholism (*Abramson et al., 2006*), and the neurophysiology of learning and memory (*Hammer & Menzel, 1995*; *Galizia, Eisenhardt & Giurfa, 2011*; *Giurfa, 2003*).

Honey bees (*Apis mellifera*) are currently the most popular species for psychological research Change to; however, bumble bees, primarily *Bombus impatiens* in North America and *B. terrestris* in Europe, have become a popular alternative due to some practical challenges related to maintaining a honey bee laboratory, such as requiring a large outdoor foraging area. While recent psychological research with bumble bees shows promising potential, one area that is notably absent from the bumble bee literature is the study of innate defensive responses, especially in conjunction with learning. For example, in honey bees, sting extension response (SER) conditioning research investigates how restrained bees learn to sting in response to a stimulus associated with shock (*Vergoz et al., 2007*; *Tedjakumala & Guirfa, 2013*). Similar work has also been conducted in other Hymenoptera (e.g., *Desmedt et al., 2017*). Unfortunately, there is not yet analogous work with bumble bees. This is surprising given that bumble bees appear to offer a unique and ideal behavior to fulfill this line of research, the disturbance leg-life response.

In this paper, we discuss the disturbance leg-lift response (DLR), and its potential use in psychological research. When exposed to a presumably threatening stimulus, bumble bees commonly react by lifting one or multiple legs (see Fig. 1). While this behavior is primarily undescribed and we have only found brief mentions in two publications (*Djegham, Verhaeghe & Rasmont, 1994*; *Free, 1958*), it appears to occur in many *Bombus* species worldwide. (Curious readers may perform an online image search for the anthropomorphizations "bumble bee high five" or "bumble bee wave"). In our first experiment, we investigate the temporal relationships between DLR, biting and stinging as an invasive stimulus approaches in order to explore potential functions of the DLR. In our second experiment, we investigate if the DLR is a suitable behavior for conditioning procedures, similar to SER. Specifically, we compare habituation of the DLR across captive and wild samples. Finally, we discuss implications for future research with special considerations for the growing use of *Bombus* species as model organisms.

## EXPERIMENT 1—THE ROLE OF DLR

In this experiment, we explore the role of DLR as a reaction to potential danger to establish an understanding of the behavior as a prerequisite to later investigations of DLR conditioning. Many species emit specific responses, like DLR, when threatened. For example, spiders may lift several front legs to reveal fangs (*Cloudsley-Thompson, 1995*), while hissing cockroaches produce an audible hiss (*Hunsinger et al., 2017*; *Shotton, 2014*). Although making distinctions between categories of antipredator responses can

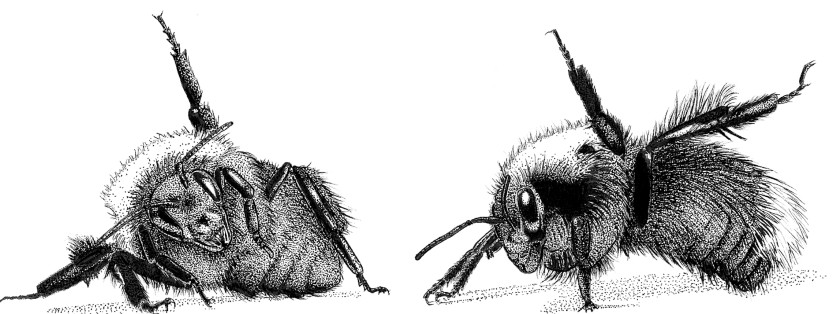

**Figure 1** **The disturbance leg-lift response (DLR) of the bumble bee.** Artwork by Jennifer Salazar. Original reference photographs by Ivan Mikhaylov.

be challenging, there are two major categories that could be considered for DLR: the aposematic display and the pursuit deterrence signal.

Conspicuous aposematic displays can signal toxicity or danger to a potential predator. The vibrant colors of poison dart frogs in the family *Dendrobatidae* illustrate a case of honest aposematic signals; the colors indicate that the frog possesses toxic alkaloid compounds (*Maan & Cummings, 2012*). Similarly, in the southern United States, the bright red banding of coral snakes (*Micrurus euryxanthus, M., fulvius,* and *M. tener*) honestly signals a potent neurotoxic venom. Several species of scarlet snake (*Cemophora sp.*) and kingsnake (*Lampropeltis sp.*) also possess similar conspicuous red banding but lack venom. For the venomless mimic snakes, the aposematic coloration is a dishonest signal (*Greene & McDiarmid, 1981*).

Pursuit deterrence signals can alert a potential predator that it has been detected, communicating vigilance and possibly fitness to the predator (*Hasson, 1991*). For example, Thomson's gazelles (*Eudorcas thomsonii*) leap vertically into the air, a behavior known as stotting (*FitzGibbon & Fanshawe, 1988*), while white-tailed deer (*Odocoileus virginianus*) erect their tails to reveal a high contrast white rump when a predator is detected (*Bildstein, 1983*), and anole lizards (*Anolis sp.*) may signal fitness to potential predators with head bobs, pushups, or dewlap extensions (*Leal & Rodríguez-Robles, 1995*). While discussions of pursuit deterrence signals typically suggest that they signal the ability to escape, it seems reasonable that they could also signal readiness to use a defense such as venom.

It is possible that the DLR of bumble bees functions in an aposematic or venom-based pursuit deterrence role. The stinging response of bumble bees and other Hymenoptera can clearly serve as the foundation for an honest warning signal, and the vibrant color patterns of many bees and wasps are one well-known aposematic display. Not only does the bright coloration lead to predators quickly learning to not consume bumble bees (*Brower, Brower & Westcott, 1960*), but this coloration also leads to mimics (*Fisher & Tuckerman, 1986*; *Plowright & Owen, 1980*). If DLR functions in either an aposematic or a venom-based pursuit deterrence role, we would expect it to be closely associated with, and precede, stinging. In the following experiment, we investigate this possibility by examining the probability and order of DLR, biting and stinging in response to invasive stimuli. If DLR

often precedes, but does not follow, stinging, this would provide the first evidence that DLR is an honest signal of envenomation potential.

## Methods

### Subjects

Captive worker bumble bees (*Bombus impatiens*, $n = 62$) collected from a single, captive-breed "Natupol" bumble bee colony purchased from Koppert Biological Systems Inc. (Howell, MI) were used as subjects. The bees were maintained in the ventilated plastic colony cage (24.5 × 21.5 × 12 cm, l × w × h) in which they were shipped. The outer cardboard layer, typically used to shield colonies from outdoor conditions, was removed except for the top piece, which ensured that the hive remained in darkness. The colony was placed on a 40-watt intellitemp heating pad (Big Apple Pet Supply; Boca Raton, FL), which maintained a temperature of about 31 °C inside the hive. The colony was connected to an adjacent empty colony cage that served as a feeding area through a clear acrylic tube (2.5 cm inner diameter). Two lights (36″ Zoo Med Reptisun T5-Ho Terrarium Hood, Zoo Med Laboratories Inc.; San Luis Obispo, CA) were placed approximately 31 cm above the colony. These light fixtures provided a full range of illumination, including ultraviolet (UV) light in the range of 280–400 nm. Bumble bees can see UV light in the range of 300–400 nm (*Skorupski & Chittka, 2010*) and naturalistic lighting conditions may be important for their growth and survival (*Blacquière, Cornelissen & Donder, 2007*).

Lights and heat were automatically turned on at 7 AM and turned off at 7 PM each day to help the bees maintain daily foraging patterns. The laboratory lacked any source of natural light, and all other lights in the laboratory were also turned off by this time. Bees were allowed ad libitum access to food (either a 50% sucrose solution (w/w) or the "Bee-happy" solution provided by Koppert Biological Systems Inc.) in the feeding cage via several paper towel wicks. Water was provided directly in the hive via syringe. A three to one mixture of pollen (Stakich Bee Pollen Powder, Stakich, Inc.; Troy, MI) and pollen substitute (Mann Lake Ultra Bee, Mann Lake LTD.; Hackensack, MN) was made available ad libitum inside the hive.

Captive worker bees were collected from the clear acrylic tube and the feeding cage, chilled in a refrigerator around 1.1 °C until inactive, then placed in the experimental apparatus. After an experimental session was complete, the bees were chilled, weighed, measured, then marked with an acrylic paint marker between the wings on the thorax before being returned to the colony. Captive bees collected and returned to the hive in this manner were observed alive and healthy up to 8 weeks after participating in an experiment. As Converse College does not require an institutional review for invertebrate research, no specific review was required for the present study.

### Procedure

Subjects were placed in individual apparatuses after being collected. Each apparatus consisted of a clear plastic cube (2.6 × 3 × 2 cm), made from a microscope cover slip container, with two holes (2.55 mm diameter) drilled on opposite sides. The size of the apparatus allowed the bees to freely move but did not permit flight or substantial relocation inside the apparatus. After being placed in the apparatus, subjects were transferred to an

experimental room, placed approximately 1.2 m apart, and allowed to acclimate for three hours.

Each bee experienced five trials with a 15-minute intertrial interval (ITI) after the acclimation period. Bees were randomly assigned to either an experimental or control group. During trials for the experimental group, a researcher startled the bee by inserting a toothpick approximately halfway (1.3 cm) into the apparatus for 10 s through the hole closest to the bee. During this time, it was possible for the bee to physically contact the toothpick. An identical procedure was used for the control bees, except that the toothpick was held outside of the apparatus adjacent to the hole. This group controlled for the general approach of the investigators, as well as the presence of a close object that could not be contacted by the bees.

During each trial, several behaviors were scored from video recordings. We recorded both the occurrence of DLR and the number of legs lifted during each DLR. We specifically defined DLR as when one or more legs were lifted above the bee, relative to the bee's position. Legs that were lifted prior to the trial were not considered a DLR; observing the movement during the trial was required to record a DLR. Biting was recorded as any time a bee visibly opened and closed its mandibles during a trial. Often bees made mandible contact with the stimulus, but this was not required. Finally, we recorded stinging any time the bee contacted the stimulus with its abdomen or directed its abdomen toward the stimulus. These abdomen curls are the first component of the sting extension response (*Gage et al., 2018*). In most cases, the stinger was obvious and contacted the stimulus. In some cases, the activity of the stinger was less clear, but the unusual abdomen curls and contact were easy to observe. These abdomen curls were only observed when stimuli were presented during the experiment or when the bees were handled during collection. We used a broad definition of stinging, relative to DLR and biting, to capture instances where the sting extension could not clearly be observed, or where the bee was not able to physically contact the stimulus from its current location. For each trial, we also recorded the order in which DLR, biting, and stinging occurred.

## Analysis

All analyses were conducted through the StatsModels package (*Perktold, Seabold & Taylor, 2018*) included in the Anaconda distribution of Python, a free scientific analysis distribution of the Python programming language (*Anaconda, 2019*; http://www.python.org). Behavior sequences (e.g., DLR then bite, or bite then DLR then sting, etc.) were analyzed with a series of repeated measures logistic regressions via generalized estimating equations (GEE; *Hardin & Hilbe, 2003*). We used this series of regressions in place of a multinomial logistic regression as GEE controls for repeated measures within subjects. This technique is also less sensitive to the need for many cases per variable than multinomial regression or chi square analyses; an important consideration for our data, as statistical comparisons between commonly and uncommonly observed behavior sequences answer important research questions. We used an interceptless model where groups are treated as two mutually exclusive variables. By default, a logistic regression's parameter estimates and associated $p$ values display a difference from a 50% binary chance level. As our sequence analysis

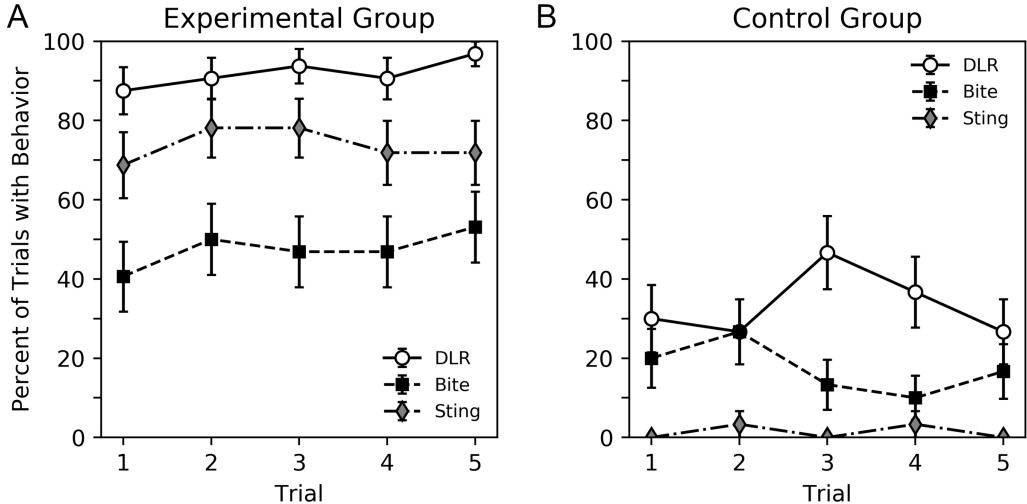

**Figure 2** **Percent of trials where bees emitted DLR, bite or sting for the experimental group (A) and control group (B).** Error bars show standard error of the mean.

considered 16 possible sequences, we subtracted the log odds of 1/16 from all parameter estimates and confidence intervals, then calculated corresponding *p* values. Each behavior sequence is therefore tested for statistical difference from chance (1/16) instead of a 1/2 comparison that is arbitrary for this data. Individual parameter estimates were compared directly by creating a z score by dividing the difference between the estimates by the square root of the sum of the squared standard errors of the estimates (*Clogg, Petkova & Haritou, 1995*; *Paternoster et al., 1998*). Even after adjusting parameters by subtracting the log odds of 1/16, the difference between estimates, z score, and *p* value are still the same as those normally reported by a regression that includes one level of a categorical variable in the intercept.

## Results

A visual overview of the findings can be seen in Figs. 2 and 3, while subsequent sections analyze our primary result, behavior sequence, in detail. Figure 2 shows the percent of trials where a DLR, bite or sting occurred for bees in the experimental and control groups. Bees in the experimental group displayed more behavior. For both groups, DLR was the most common behavior. Bees in the experimental group were more likely to sting than bite, while bees in the control group were more likely to bite than sting. The average number of legs lifted in trials where DLR occurred can be seen in Fig. 3. Not only were bees in the experimental group more likely to emit a DLR (Fig. 2), but they also lifted more legs on average. Both Figs. 2 and 3 show little change across trial that would suggest habituation, sensitization, or fatigue. While our analysis in subsequent paragraphs focuses on behavior sequences, we also included our initial exploratory analysis of individual behaviors (without considering their order) in supplementary material that will relate well to Figs. 2 and 3.

Table 1 shows the percent of trials where a particular behavior sequence occurred. For each trial, the order of DLR, bite and sting were recorded, resulting in 16 possible

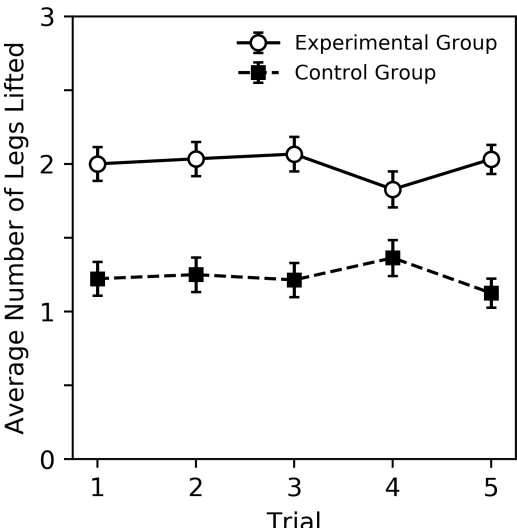

**Figure 3** **Average number of legs lifted during trials where DLR occurred for the experimental and control groups.** Error bars show standard error of the mean.

sequences (no tied rankings were observed). Corresponding statistical analysis that show the log odds of each sequence compared to a 1/16 chance value can be seen in Tables 2 and 3, with Table 2 displaying information for the experimental group, and Table 3 displaying information for the control group. We only included the groups as parameters in this analysis given the large number of behavior sequences to be analyzed and the lack of a trial effect in previous graphs. The direction and magnitude of the parameter estimates, as well as the $p$ values, reflect a difference from the chance value. Although we used separate tables due to the large size of a single combined table, the sequence analyses presented on Tables 2 and 3 should be interpreted together.

Bees in the experimental group were highly active, only being inactive 3% of the time. DLR was emitted first or by itself around 81% of trials, while biting and stinging rarely occurred first or by themselves. All the DLR-first sequences, except for DLR:Bite, occurred significantly more than chance ($p$ values < 0.000). The least common of the DLR-first sequences was the DLR:Bite sequence, occurring during only 1% of trials, and was the only DLR-first sequence to occur significantly less than chance ($p = 0.017$). Trials with a sting following DLR were much more common (30%), and trials with a bite and sting following DLR (in either order) were also more common (31%). Taken together, this indicates that in the experimental group, DLR is more related to subsequent stings than to subsequent bites. Bees in the control group were inactive during 51% of trials, significantly more than chance ($p <$ 0.000). During trials when bees were active, DLR often occurred by itself (31%), or a bite occurred by itself (15%). These were the only behaviors that occurred significantly more than chance ($p$ values < 0.002). Sequences of multiple behaviors were rare.

Table 4 shows pairwise comparisons between DLR:Bite and other DLR-first sequences, as well as comparisons between DLR:Sting and other DLR-first sequences for the experimental

**Table 1  Percent of trials with behavior sequence.**

| Behavior Sequence | Experimental | Control |
|---|---|---|
| Inactive | 3.12 | 51.33 |
| DLR | 18.12 | 30.67 |
| DLR:Bite | 1.25 | 0.00 |
| DLR:Sting | 30.00 | 0.67 |
| DLR:Bite:Sting | 16.25 | 0.00 |
| DLR:Sting:Bite | 15.62 | 0.00 |
| Bite | 2.50 | 14.67 |
| Bite:DLR | 1.25 | 2.00 |
| Bite:Sting | 1.25 | 0.67 |
| Bite:DLR:Sting | 5.62 | 0.00 |
| Bite:Sting:DLR | 2.50 | 0.00 |
| Sting | 0.62 | 0.00 |
| Sting:DLR | 0.62 | 0.00 |
| Sting:Bite | 0.62 | 0.00 |
| Sting:DLR:Bite | 0.00 | 0.00 |
| Sting:Bite:DLR | 0.62 | 0.00 |

**Table 2  Experimental group sequence regression.**

| Sequence | Estimate | Standard error | 95% Confidence intervals | | $p$ value |
|---|---|---|---|---|---|
| Inactive | −0.726 | 0.424 | −1.557 | 0.105 | 0.087 |
| DLR | 1.200 | 0.306 | 0.601 | 1.799 | 0.000 |
| DLR:Bite | −1.661 | 0.693 | −3.020 | −0.303 | 0.017 |
| DLR:Sting | 1.861 | 0.183 | 1.501 | 2.220 | 0.000 |
| DLR:Bite:Sting | 1.068 | 0.229 | 0.619 | 1.517 | 0.000 |
| DLR:Sting:Bite | 1.022 | 0.266 | 0.500 | 1.543 | 0.000 |
| Bite | −0.956 | 0.601 | −2.134 | 0.223 | 0.112 |
| Bite:DLR | −1.661 | 0.693 | −3.020 | −0.303 | 0.017 |
| Bite:Sting | −1.661 | 0.693 | −3.020 | −0.303 | 0.017 |
| Bite:DLR:Sting | −0.112 | 0.343 | −0.784 | 0.560 | 0.744 |
| Bite:Sting:DLR | −0.956 | 0.601 | −2.134 | 0.223 | 0.112 |
| Sting | −2.361 | 0.990 | −4.302 | −0.420 | 0.017 |
| Sting:DLR | −2.361 | 0.990 | −4.302 | −0.420 | 0.017 |
| Sting:Bite | −2.361 | 0.990 | −4.302 | −0.420 | 0.017 |
| Sting:DLR:Bite | −16.495 | 0.177 | −16.841 | −16.148 | 0.000 |
| Sting:Bite:DLR | −2.361 | 0.990 | −4.302 | −0.420 | 0.017 |

group. The estimate differences and z scores were calculated from the parameter estimates and standard errors reported in Tables 2 and 3. The pairwise comparisons show that the DLR:Bite sequence occurs significantly less than all other DLR-first sequences ($p$ values < 0.000). Conversely, the DLR:Sting sequence occurs significantly more than DLR:Bite, and other DLR-first sequences ($p$ values < 0.009) except for DLR alone. Though DLR:Sting does occur more than DLR alone, the difference is not significant ($p = 0.064$). Note that for

**Table 3  Control group sequence regression.**

| Sequence | Estimate | Standard error | 95% Confidence Intervals | | p value |
|---|---|---|---|---|---|
| Inactive | 2.761 | 0.247 | 2.278 | 3.245 | 0.000 |
| DLR | 1.892 | 0.269 | 1.366 | 2.419 | 0.000 |
| DLR:Bite | −17.495 | 0.183 | −17.853 | −17.137 | 0.000 |
| DLR:Sting | −2.296 | 0.990 | −4.236 | −0.356 | 0.020 |
| DLR:Bite:Sting | −17.495 | 0.183 | −17.853 | −17.137 | 0.000 |
| DLR:Sting:Bite | −17.495 | 0.183 | −17.853 | −17.137 | 0.000 |
| Bite | 0.947 | 0.291 | 0.377 | 1.518 | 0.001 |
| Bite:DLR | −1.184 | 0.559 | −2.279 | −0.088 | 0.034 |
| Bite:Sting | −2.296 | 0.990 | −4.236 | −0.356 | 0.020 |
| Bite:DLR:Sting | −17.495 | 0.183 | −17.853 | −17.137 | 0.000 |
| Bite:Sting:DLR | −17.495 | 0.183 | −17.853 | −17.137 | 0.000 |
| Sting | −17.495 | 0.183 | −17.853 | −17.137 | 0.000 |
| Sting:DLR | −17.495 | 0.183 | −17.853 | −17.137 | 0.000 |
| Sting:Bite | −17.495 | 0.183 | −17.853 | −17.137 | 0.000 |
| Sting:DLR:Bite | −16.495 | 0.183 | −16.853 | −16.137 | 0.000 |
| Sting:Bite:DLR | −17.495 | 0.183 | −17.853 | −17.137 | 0.000 |

**Table 4  Experimental group DLR-first pairwise comparisons.**

| Comparison | Estimate difference | z score | p value |
|---|---|---|---|
| DLR:Bite vs. DLR | −2.862 | −3.776 | 0.000 |
| DLR:Bite vs. DLR:Sting | −3.522 | −4.911 | 0.000 |
| DLR:Bite vs. DLR:Bite:Sting | −2.730 | −3.738 | 0.000 |
| DLR:Bite vs. DLR:Sting:Bite | −2.683 | −3.613 | 0.000 |
| DLR:Sting vs. DLR | 0.661 | 1.852 | 0.064 |
| DLR:Sting vs. DLR:Bite:Sting | 0.792 | 2.700 | 0.007 |
| DLR:Sting vs. DLR:Sting:Bite | 0.839 | 2.597 | 0.009 |

this series for pairwise comparisons, it may be appropriate to use a multiple comparison correction to adjust the significance threshold. For example, the conservative Bonferroni correction would involve dividing the alpha level by the number of comparisons, in this case 0.05 divided by 7 produces a new significance threshold of 0.007, which may affect interpretation of the comparisons between DLR:Sting and DLR-first sequences containing both a sting and a bite. The reader is free to use whichever correction technique they deem appropriate.

Taken together, the findings reported in Tables 1–4 strongly suggest that DLR is more related to subsequent stinging than it is to subsequent biting for the experimental group. This is also in line with our supplementary analysis of individual behaviors. Ultimately, the fact that DLR often precedes stinging in the experimental group, the group where stinging frequently occurred, indicates DLR may function to signal potential predators that a sting is imminent.

## Discussion

Our findings describe DLR, a primarily undocumented behavior, and show that it often precedes stinging, but rarely precedes biting alone. We also demonstrate that the probability of DLR is sensitive to stimulus intensity, as the increased proximity to the stimulus in the experimental group, compared to the control group, altered behavior. Together, these results suggest that DLR is an honest signal that indicates stinging may occur. DLR may function in either an aposematic or a pursuit deterrence role, and these functions may not be mutually exclusive. Given the already bright coloration of bumble bees, it is possible that DLR serves as a multimodal enhancement of existing aposematic signals, adding a conspicuous posture to vibrant colors (for discussions on multimodal antipredator signals see *Ritson-Williams & Paul, 2007*; *Rowe & Guilford, 1999*; *Rowe & Haplin, 2013*). If DLR has a pursuit deterrence function, it may signal that the bee is aware of a potential predator and will sting if pursued.

While DLR likely signals a sting may occur, DLR can also occur alone. In the control group, bees emit DLR but rarely sting. This likely occurs because the distant stimulus is intense enough to elicit DLR, but does not support stinging. In the wild, if DLR is successful at preventing a potential predator attack, it may occur without subsequent stinging behavior. Therefore, the occurrence of DLR in the control group is consistent with an honest signal interpretation.

While our experiment clearly indicates a temporal connection between DLR and stinging, additional research is needed to clarify DLR's specific antipredator function. Such research will need to consider what stimuli and predators elicit DLR, and equally importantly, how predators respond. Field experiments may also study DLR in situ, providing bees with a number of alternative behaviors, including fleeing. Such ecologically valid research may be required to completely determine the function of DLR. For example, if DLR functions strictly as a pursuit deterrence signal, bees may emit a DLR, then flee if a predator approaches, while if DLR has only an aposematic function it may not be related to any antipredator behavior other than stinging.

In addition to further clarifying the function of DLR, our initial work facilitates many additional research topics. For example, research may consider how DLR relates to specific stimulus modalities or intensities, and if predators have learned or innate responses to DLR. Research should also consider the extent that DLR occurs in other *Bombus* species, and if it differs across species. Studies of individual differences will likely also be fruitful, especially considering recent literature on the size-dependent behavior in bumble bee workers (e.g., *Jandt, Huang & Dornhaus, 2009*; *Kodaira et al., 2009*; *Raine & Chittka, 2008*; *Riveros & Gronenberg, 2009a*; *Riveros & Gronenberg, 2009b*; *Spaethe & Weidenmüller, 2002*). Finally, as bees are social animals, future work should also consider social factors, such as if DLR can be elicited by other bees or if DLR affects adjacent bees. It may also be possible that DLR, like a number of other behaviors, is affected by alarm pheromones (e.g., *Avalos et al., 2017*; *Rossi, D'Ettorre & Giurfa, 2018*). Given that bumble bees possess tarsal glands (*Pouvreau, 1991*; *Schmitt, 1990*), social odors may even be released during DLR.

## EXPERIMENT 2—HABITUATION OF DLR

In this experiment, we investigated the possibility that DLR could change as a function of learning. Specifically, we wanted to know if DLR habituates to repeated mild stimuli. Habituation, defined as the diminishing of a response, emotional or physical, to a repeated stimulus (*Thompson & Spencer, 1966*), is a simple form of learning that can be observed across nearly all species, from planarians (*Nicolas, Abramson & Levin, 2008*) to rodents (*Geyer & Braff, 1987*). Habituation of disturbance responses has also been documented in many species. For example, hissing cockroaches may cease emitting their disturbance hiss in the presence of specific handlers (*Davis & Heslop, 2004*), rattlesnakes show a reduction in latency and duration of rattling in response to a startling stimulus (*Place & Abramson, 2008*), and the gill withdrawal reflex of the sea hare *Aplysia* is also known to habituate (*Carew, Pinsker & Kandel, 1972*). Studies of habituation are also often the foundation for other procedures, including investigations of mental health (*Akdag et al., 2003*; *Geyer & Braff, 1987*; *Jaycox, Foa & Morral, 1998*), and neurological processes related to learning and memory (*Castellucci & Kandel, 1974*; *Castellucci et al., 1970*). If DLR can be altered through habituation learning, this opens new possibilities in behavioral and physiological research with bumble bees.

In addition to discovering if DLR can change as a function of learning, we were also interested in differences across populations due to differences we observed in pilot research. Specifically, we compared samples of captive bred to wild caught bumble bees. Given the substantial number of findings on behavioral differences in honey bees due to breed, genetics and environment (e.g., *Alaux et al., 2009*; *Schulz, Haung & Robinson, 1998*; *Sheppard et al., 1997*; *Spivak, 1997*; *Tautz et al., 2003*), it is reasonable to investigate if some differences may be found between captive bred and wild caught bumble bees. If DLR and habituation of DLR are observed across samples of both populations, this would also suggest that DLR may be a robust behavior to study in learning experiments. This would be a beneficial comparison, considering the use of both wild and captive bees in the literature.

### Methods
#### Subjects

Both captive ($n = 64$) and wild worker bumble bees ($n = 64$) were used in this experiment. Captive bees were acquired and maintained in a similar manner as described in experiment 1 with a few exceptions. The captive colony was connected to a screen flight cage ($91 \times 46 \times 46$ cm, l × w × h), made from a modified Zoo Med "Reptibreeze" reptile cage via a 16 cm long, clear acrylic tube (2.5 cm inner diameter). In this flight cage, bees fed from plastic dishes. Two lights (one 36″ Zoo Med Reptisun T5-Ho Terrarium Hood and one 30–38″ Zoo Med Reptisun LED Terrarium Hood) were placed approximately 31 cm above the colony, providing a full range of light.

Captive bees were collected in the flight cage and prepared in a similar manner as described in the previous experiment and were observed alive and healthy up to 4.5 weeks after participating in an experiment. Thirty-two captive bees were collected from one colony in 2018, while the remaining 32 captive bees were collected from a second colony in 2020. Wild worker bees were collected while foraging, primarily on clover and *Abelia*, at the

Converse College campus (Spartanburg, SC). Thirty-two wild bees were collected during July 2018, while the remaining 32 wild bees were collected during August and September 2020. Procedures for capturing, chilling, using, and marking wild bees were similar to the procedures for captive bees. Wild bees were released at the capture location, and many immediately returned to foraging. Marked wild bees were observed foraging two weeks after the experiment. As Converse College does not require an institutional review for research with non-threatened invertebrates, no specific review or permits were required for the present study.

Captive bees were visibly smaller than their wild counterparts. We recorded head width in a sample of 249 bees; 123 captive and 62 wild bees from 2018 pilot research, as well as the 32 captive and 32 wild bees collected for this experiment in 2020. An independent samples $t$-test revealed that captive bees were significantly smaller that wild bees (mean difference $= -0.61$ mm, $t_{247} = -10.727$, $p < 0.000$).

## Procedure

Subjects were placed in individual apparatuses after being collected. Each apparatus consisted of a capsule formed from a clear acrylic tube (4.5 cm long, 2.5 cm inner diameter), with two white plastic caps sealing the tube. Two holes (0.4 cm diameter) were drilled near the center of each cap. Each apparatus was placed approximately half a meter apart, and bees were allowed an acclimation period of 45 min.

Each bee experienced 10 trials with a seven-minute ITI after the acclimation period was complete. During each trial, a researcher startled the bee by presenting a hand 15 cm above the apparatus, rapidly lowering it to approximately 6 cm above the apparatus, rotating the hand once in a clockwise circular motion, and then withdrawing the hand. As strong stimuli can inhibit habituation or cause sensitization, we used this relatively mild stimulus, compared to those used in the first experiment, to increase the chance that habituation could be observed. The bees' response was recorded during the two-second stimulus presentation and for three seconds after the presentation. DLR was recorded as a binary response and no other behaviors were recorded. Subjects collected in 2020 also experienced three additional trials. On the 11th trial, the bottom of the apparatus was tapped once, out of view of the subject. This trial served as a dishabituation trial to determine if DLR would occur to other stimuli and verify that any decrease in DLR during the preceding 10 habituation trials was not caused by fatigue. The 12th trial was a return to the standard habituation trial. Finally, on the 13th trial, subjects were collected by reaching directly toward the front of the apparatus, visible to the subject, and holding a hand adjacent to, but not touching, the apparatus for the five-second observation period before placing the apparatus in the refrigerator for chilling. All bees, in both 2018 and 2020, were chilled after recollection, marked, and returned to their colony or the collection area.

## Analysis

We analyzed the probability of DLR across trial using repeated measures logistic regression via GEE. As with previous logistic regressions, we use an interceptless form so that the parameters can be directly compared to a chance value (in this case 50%), then compared

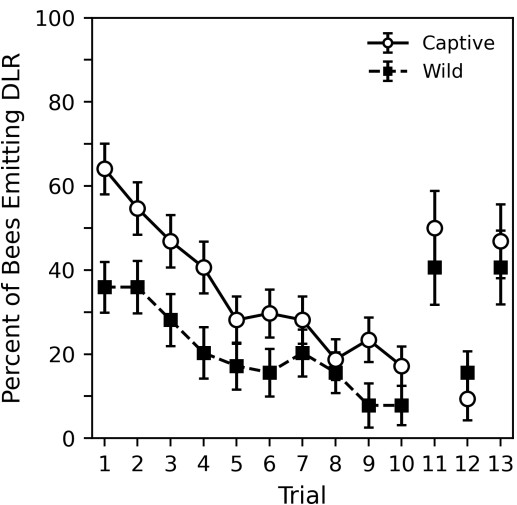

**Figure 4** **Percent of captive and wild bees emitting DLR across the 10 habituation trials and final three control trials.** Sixty-four subjects per sample were used in trials 1 to 10. The final three trials represent 32 subjects per sample. Error bars show standard error of the mean.

them to each other by creating a z score by dividing the difference between the estimates by the square root of the sum of the squared standard errors of the estimates. As initial analysis revealed no significant differences between bees in the 2018 and 2020 collection periods ($p$ values < 0.489), and there was no theoretical reason to expect differences, we did not include collection period as a parameter in our main analysis.

## Results

Figure 4 shows the percent of captive and wild bees emitting DLR across the 10 habituation trials in both the 2018 and 2020 collection periods while Table 5 shows corresponding statistical analysis. The captive bees were initially much more likely to respond; nearly 65% of captive bees responded compared to around 35% of wild bees. The analysis shows that the initial probability of response for captive bees was significantly greater than chance ($p = 0.005$). Wild bees initially responded less than chance, but not significantly so ($p = 0.157$). A direct comparison of the parameters revealed that captive bees were significantly more likely to initially respond than the wild bees (estimate difference = 0.931, $z = 2.899$, $p = 0.004$). The probability of response decreased significantly across trials for both captive and wild bees at a somewhat similar rate ($p$ values < 0.000). Though the captive bees showed a slightly stronger effect, a direct comparison reveals that this difference was not significant (estimate difference = $-0.030$, $z = -0.549$, $p = 0.583$).

The final three trials, also shown in Fig. 4, were only conducted with subjects collected in 2020. Corresponding analysis comparing the final three trials to trials 1 and 10 can be seen in Table 6. While the original GEE regression is shown in supplemental material (Table S4), it is more useful to interpret the pairwise comparisons shown in Table 6 that were derived from the GEE regression. The 11th dishabituation trial shows the probability of response returned to nearly a trial 1 level, as does the 13th collection trial. The analysis shows that

**Table 5 Change in DLR across trial.**

| Parameter | Estimate | Standard error | 95% Confidence Intervals | | p value |
|---|---|---|---|---|---|
| Captive | 0.587 | 0.210 | 0.177 | 0.998 | 0.005 |
| Wild | −0.344 | 0.243 | −0.821 | 0.132 | 0.157 |
| Captive * Trial | −0.232 | 0.035 | −0.301 | −0.163 | 0.000 |
| Wild * Trial | −0.202 | 0.042 | −0.284 | −0.120 | 0.000 |

for both captive and wild bees, these response levels are statistically indistinct ($p$ values > 0.317). The 12th trial was a return to a standard habituation trial and approximately follows the trends seen in the previous 10 habituation trials. The analysis shows that, although the response level for wild bees on the 12th trial was somewhat higher than might be expected, for both captive and wild bees, these response levels are statistically indistinct ($p$ values > 0.453). Both trial 10, the final habituation trial, and trial 12, the return to habituation trial, showed significantly lower probabilities of response than trial 11, the dishabituation trial, and trial 12, the collection trial ($p$ values < 0.032). As with the previous pairwise comparisons in experiment 1, it may be appropriate to use a multiple comparison correction to adjust the significance threshold. In this case, interpretations of comparisons between the wild bees' probability of response on trial 12 may change slightly. Overall, the tendency of bees to perform DLR when stimuli were presented in the 11th and 13th, trials suggests that the decrease in DLR during the habituation trials occurred due to habituation learning, not motor fatigue or sensory adaptation.

## Discussion

In this experiment, we provided the first demonstration of habituation of DLR, as well as documented differences in DLR across samples of captive and wild populations. While both samples showed a similar rate of habituation, the captive bees were initially more likely to perform DLR. This difference in DLR may have occurred for two different reasons. First, the samples of worker bees we collected from captive and wild populations may have differed in role specialization. Bumble bee castes include the reproductive queen and drone castes, as well as the primarily non-reproductive worker caste frequently used in research. Workers may be further specialized. The smaller worker bees are more likely to feed larvae and attend to hive maintenance, while larger workers act as foragers. In bumble bees, role specialization appears to be determined during early development, and research suggests that physical dimensions can predict behavioral performance (*Jandt, Huang & Dornhaus, 2009*; *Kodaira et al., 2009*; *Raine & Chittka, 2008*; *Riveros & Gronenberg, 2009a*; *Riveros & Gronenberg, 2009b*; *Spaethe & Weidenmüller, 2002*).

The bees sampled from our captive population were significantly smaller than those from our sample of wild bees, suggesting the wild bees were more likely to be foragers. The wild bees were also clearly collected during the act of foraging, further increasing our confidence they fit this role specialization. While our captive bees were also collected in their foraging area, we are less confident this collection method ensures they are true foragers due to one of the challenges of maintaining an indoor bumble bee hive. In laboratory colonies, bumble bee workers may not return to the hive after feeding, and may

**Table 6  Pairwise comparisons of trials.**

| Comparison | Difference | z score | p value |
|---|---|---|---|
| Captive 1 vs. Captive 10 | 2.325 | 3.609 | 0.000 |
| Captive 1 vs. Captive 11 | 0.379 | 0.752 | 0.452 |
| Captive 1 vs. Captive 12 | 2.648 | 3.755 | 0.000 |
| Captive 1 vs. Captive 13 | 0.505 | 0.999 | 0.318 |
| Captive 10 vs. Captive 11 | −1.946 | −3.036 | 0.002 |
| Captive 10 vs. Captive 12 | 0.323 | 0.399 | 0.690 |
| Captive 10 vs. Captive 13 | −1.821 | −2.839 | 0.005 |
| Captive 11 vs. Captive 12 | 2.269 | 3.232 | 0.001 |
| Captive 11 vs. Captive 13 | 0.125 | 0.250 | 0.803 |
| Captive 12 vs. Captive 13 | −2.144 | −3.052 | 0.002 |
| Wild 1 vs. Wild 10 | 2.325 | 3.609 | 0.000 |
| Wild 1 vs. Wild 11 | 0.379 | 0.752 | 0.452 |
| Wild 1 vs. Wild 12 | 2.648 | 3.755 | 0.000 |
| Wild 1 vs. Wild 13 | 0.505 | 0.999 | 0.318 |
| Wild 10 vs. Wild 11 | −1.946 | −3.036 | 0.002 |
| Wild 10 vs. Wild 12 | 0.323 | 0.399 | 0.690 |
| Wild 10 vs. Wild 13 | −1.821 | −2.839 | 0.005 |
| Wild 11 vs. Wild 12 | 2.269 | 3.232 | 0.001 |
| Wild 11 vs. Wild 13 | 0.125 | 0.250 | 0.803 |
| Wild 12 vs. Wild 13 | −2.144 | −3.052 | 0.002 |

instead inhabit a flight cage or feeding area. Workers may build clusters of cells near a food dish, store food, and even raise drones. We call this tendency the "lost bee effect." If not carefully managed, this may result in half the colony moving away from the hive within a month (see Fig. S1 in the supplemental material). Although not widely published (we have only seen this reported in *Jandt & Dornhaus, 2009* and *Blacquière, Cornelissen & Donder, 2007*), the lost bee effect appears to be a common issue. While there is no clear solution, preventative measures include capturing lost bees each day, returning them to the colony, and cleaning the flight cage to remove odors (F Muth, pers. comm., 2018); capturing and returning lost bees while also killing repeat offenders (J Jandt, pers. comm., 2018); feeding the bees inside the hive and completely preventing access to other areas (W Gronenberg & A Riveros, pers. comm., 2018); and providing a smaller antechamber between the hive and the flight cage to encourage lost bees to build closer to the hive (J Nieh, pers comm., 2018). While our observations and the above communications are in regard to the North American *Bombus impatiens*, the lost bee effect also occurs in the European *B. terrestris* (L Chittka, pers. comm., 2018). Given the tendency of worker bees to become "lost", it may not be possible to ensure that bees collected near food are the larger, foraging-specialized workers. Instead, collected subjects may also consist of smaller bees specialized for brood care and hive maintenance. While a growing body of research demonstrates the usefulness of laboratory colonies, the lost bee effect indicates they may not be thriving, and this may impact their use as model organisms. In our case, the lost bee effect may have made our sample of captive bees less likely to be foragers.

A second possible reason for the difference in DLR between captive and wild bees may be the distinct experiences of bees raised indoors compared to those of wild bees. Captive bees were only exposed to stimuli in their hive and flight cage, and ultimately experienced only a small number of stimuli before research. Conversely, wild bees likely contact many stimuli during daily foraging including other insects, birds, pedestrians, and even landscaping equipment. It is possible that exposure to a wide variety of stimuli served to acclimate the wild bees to mild visual stimuli, such as the hand wave used in this experiment.

Regardless of the difference in initial rate of DLR, both captive and wild bees showed clear habituation trends, and thus our experiment suggests expansive opportunities for a new area of non-associative learning research with bumble bees. Future research may consider the principles of habituation and sensitization outlined by *Thompson & Spencer (1966)*, *Groves & Thompson (1970)*, and *Rankin et al. (2009)*. For example, altering the time between stimulus presentations may change the rate of habituation, and placing the animal in an agitated state prior to habituation trials may instead result in sensitization. Future work may also consider exploring classical conditioning or operant conditioning of DLR. This would be a reasonable next step considering the reports of sting extension response (SER) conditioning in honey bees, though the primary author and one reviewer note that SER conditioning may be less robust than the literature suggests. Additionally, various DLR conditioning studies could be used as a basis for research on pesticides, sensory perception, memory, pharmacology, and neurophysiology research, as conditioning research with honey bees has also done for these same topics (e.g., *Abramson et al., 2004*; *Abramson et al., 2006*; *Faber, Joerges & Menzel, 1999*; *Giurfa et al., 2009*; *Linader, De Ibrra & Laska, 2012*; *Mustard et al., 2012*; *Varnon et al., 2018*; *Vergoz et al., 2007*).

## CONCLUSIONS

Our experiments document a primarily undescribed behavior, the disturbance leg-lift response (DLR). We suggest an antipredation role for DLR, show that DLR can change as a function of learning, and outline future considerations for DLR as a behavior of interest for both behavioral ecology and comparative psychology. A growing body of research with bumble bees is indicating they are becoming an important model organism for ecological, behavioral, and physiological research. We hope that our work will stimulate additional research on DLR, and on bumble bees in general. We also hope that special considerations will be given to reporting not only what bumble bees can do, but also what they cannot do. Reporting differences, including deficits, is an important component of research in animal behavior (*Avarguès-Weber & Giurfa, 2013*), and this is especially important for new model organisms.

### Funding

This project was supported by grant P20GM103499 (SC INBRE) from the National Institute of General Medical Sciences, National Institutes of Health. The funders had no role in study design, data collection and analysis, decision to publish, or preparation of the manuscript.

### Grant Disclosures

The following grant information was disclosed by the authors:
National Institute of General Medical Sciences: P20GM103499.
National Institutes of Health.

### Competing Interests

The authors declare there are no competing interests.

### Author Contributions

- Christopher A. Varnon conceived and designed the experiments, analyzed the data, prepared figures and/or tables, authored or reviewed drafts of the paper, and approved the final draft.
- Noelle Vallely, Charlie Beheler and Claudia Coffin conceived and designed the experiments, performed the experiments, authored or reviewed drafts of the paper, and approved the final draft.

### Data Availability

Raw data is available in the Supplemental Files.

### Supplemental Information

Supplemental information for this article can be found online at http://dx.doi.org/10.7717/peerj.10997#supplemental-information.

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
