# Peer review of "The disturbance leg-lift response (DLR): an undescribed behavior in bumble bees"

_PeerJ, doi:10.7717/peerj.10997_

## Round 0.1 · original submission · Major Revisions

· Academic Editor

Major Revisions

The reviewers have much appreciated your work. Yet, both underline a major issue about the habituation experiment, and I agree with them.

·

Basic reporting

Few typos to be corrected (see yellow and red corrections in the annotated PDF file).

The literature references in the introduction seem a little bit biased (see annotated PDF file for suggestions).

The structure of the article is somewhat unorthodox (each experiment was described and discussed separately), especially with regard to the structure of the results section where each paragraph corresponds to a figure or table. I think that reading the manuscript would benefit from focusing the writing on the story told and the main results instead of dissecting each element of the display separately when they clearly relate to the same result. In addition, when a result is stated in a sentence, the statistics should be reported in the same sentence to support that statement. In my opinion, it is not necessary to describe statistical tables separately from descriptive figures/tables (see annotated PDF file).

Experimental design

Be careful not to oversell your study, the DLR has already been described. Nevertheless, it is true that it has not been specifically studied, especially not in the context of learning (see the annotated PDF file for suggestions).

Controls are lacking to be certain of your interpretation of the bees' behavior as being the result of habituation (see annotated PDF file).

I missed a few details regarding the apparatus in Experiment 1 (see annotated PDF file).

Validity of the findings

I would advise to be more cautious regarding your conclusions on habituation, given the lack of appropriate controls (see annotated PDF file).

Reviewer 2 ·

Basic reporting

No comment. Please see in "General comments for the author"

Experimental design

No comment. Please see in "General comments for the author"

Validity of the findings

No comment. Please see in "General comments for the author"

Additional comments

In the paper titled “The disturbance leg-lift response (DLR): An undescribed behavior in bumble bees” Varnon and collaborators described and investigated the function of an undocumented behavior, the disturbance leg-lift response (DLR) that bumblebees (Bombus impatiens) showed after disturbance. In particular, in a first part of the study, they tried to induce visual and mechanical disturbance to focal bees and recorded the occurrence and sequence of DLR, biting, and stinging. In a second part of the study, after having characterized the DLR response, they investigated if DLR could be altered by non-associative learning in a habituation assay. The authors concluded that DLR may function as a warning signal that a sting will occur and that DLR can be used in non-associative learning test.
Research questions are well defined, relevant, and meaningful. The paper is timely, original, and potentially useful for future studies, particularly on aversive learning and memory. The paper is clear. Structure conforms to PeerJ standards. Figures are all relevant and sufficiently described. As far as I can understand, the results are statistically sound. Methods are described with sufficient detail (but see some minor points). The work potentially worth publishing. I have however, a main concern and several minor points that the authors should address prior to the publication of the manuscript. I have listed below all these points hoping that my comments will be helpful for revising the work.

Main point
My most significant concern about the manuscript is that the second experiments failed in demonstrating the occurrence of a true habituation of DLR. After presenting the bumblebees with the habituation stimulus several times, the authors correctly induced dishabituation by recollecting them from the apparatus. This recollection served as dishabituating stimulus. However, I am afraid, the authors missed the final step to demonstrate that what they observed in the habitation assay was a case of true habituation. That is, after dishabituation, the bees should had been presented once again with the original habituation stimulus and observe the recovery from habituation in a very precise manner. Without this crucial final step, they authors cannot rule out fatigue or sensory adaptation. The authors seemed to be aware of this as in line 376-377 they said that they “informally observed 50% of the bees displaying DLR” when moved. At the very end of the habituation trial about the 25% and 20% of wild and captive bees displayed DLR (fig. 4). Whether this is statistically different from the vague and too approximate 50% is not known.

Minor points
Line 49-57 the entire paragraph needs to be improved a bit. Instead of making a list of learning/cognitive tasks/abilities, some of which are too narrow and punctiform examples (such as perception of time, learned helplessness, select and reject stimulus control) the authors might rather mentioned few important reviews or books (For Chittka e Thomson 2001; Giurfa 2003, 2013; Galizia and Eisenhardt 2011). The literature should be better referenced.
Lines 58-61. I respectfully disagree. Bumblebees are a model system since many years. More importantly, I do not think these lines are needed.
Line 63 Authors mentioned the existence of several aversive conditioning protocols. They mentioned the sting extension response (SER) for honeybees. I guess it would be very relevant to mention others such as the aversive conditioning of the mandible opening response in ants (MOR, Desment et al 2017) or the aversive conditioning in mosquito larvae (Banglan et al 2007).
Line 74 full dot is missing.
Line 79 it is not clear to me why the authors decided to compare the DLR in wild and captive bees. Please explain why you used this method.
Line 159 I guess 1.3 mm should be 1.3 centimetres. Please double check this
Line 177 What does it mean “context” here? Please try to be a bit clearer.
Line 261. I might be wrong here, but how the authors inferred that “DLR is sensitive to stimulus intensity, changing as stimuli become closer across groups” if the toothpick was always kept 12 mm apart from the body of the bumblebees in the experimental group? Please clarify this point.
Line 326. Is there any particular reason why each bee experienced 10 trials with a seven-minute ITI. Why seven? did you try other combinations in pilot study?
Line 375 Please report full stats
Line 284 Please report full stats
Line 388 The authors mentioned two populations. Yet, as far as I understand, they tested one single wild colony and one single reared colony. This is a confounding factor as one needs more replicates to say that the observed difference is due to rearing condition (wild vs captive) and not colony differences.

Line 402 how did you collect captive bees? Please clarify. The fact of being or not a forager might be even more crucial than being big or small in displaying a DLR. Collecting foragers at a feeder may have solved this problem.

---

## Round 0.2 · Minor Revisions

· Academic Editor

Minor Revisions

Thank you very much for improving the manuscript. As indicated by reviewer 1, there remain some minor points needing your attention.

·

Basic reporting

no comment

Experimental design

no comment

Validity of the findings

no comment

Additional comments

The manuscript reads well and the results of the added trials to demonstrate habituation are convincing.
I do not have any major comment, only minor ones (see below):
Line 70: “Desment et al. 2017” should read “Desmedt et al. 2017”.
Lines 214-216: I believe that the authors have forgotten to address my concern about the lack of reported statistical results to support such statements in my first review.
Line 254: I would report the p-values here, i.e. (p values < 0.009), for the comparison between the DLR:Sting sequence and other DLR-first sequences.
Line 268: “an primarily undocumented…” should read “a primarily undocumented…”.
Line 296: “innate response” should read “innate responses”.
Line 379: “proceeding” should read “preceding”.
Line 392: commas missing.
Line 407: Replace “Figure 3” by “Figure 4”.
Supplementary line 86: Replace “DLR are sting” by “DLR and sting”.
Suppl. Line 124: Delete “the from” before “their hive”, and “with a similar main colony”.

Reviewer 2 ·

Basic reporting

no comment

Experimental design

no comment

Validity of the findings

no comment

---

## Round 0.3 · accepted · Accept

· Academic Editor

Accept

Thank you very much for the improvements introduced in the new version of the manuscript.